# Interaction of TOR and PKA Signaling in *S. cerevisiae*

**DOI:** 10.3390/biom12020210

**Published:** 2022-01-26

**Authors:** Michael Plank

**Affiliations:** Department of Molecular and Cellular Biology, University of Arizona, Tucson, AZ 85721-0206, USA; plank@email.arizona.edu

**Keywords:** TOR, PKA, signaling pathway interaction, kinase, substrate specificity, cross-talk, ribosome production, stress response, autophagy, nutrient sensing

## Abstract

TOR and PKA signaling are the major growth-regulatory nutrient-sensing pathways in *S. cerevisiae*. A number of experimental findings demonstrated a close relationship between these pathways: Both are responsive to glucose availability. Both regulate ribosome production on the transcriptional level and repress autophagy and the cellular stress response. Sch9, a major downstream effector of TORC1 presumably shares its kinase consensus motif with PKA, and genetic rescue and synthetic defects between PKA and Sch9 have been known for a long time. Further, studies in the first decade of this century have suggested direct regulation of PKA by TORC1. Nonetheless, the contribution of a potential direct cross-talk vs. potential sharing of targets between the pathways has still not been completely resolved. What is more, other findings have in contrast highlighted an antagonistic relationship between the two pathways. In this review, I explore the association between TOR and PKA signaling, mainly by focusing on proteins that are commonly referred to as shared TOR and PKA targets. Most of these proteins are transcription factors which to a large part explain the major transcriptional responses elicited by TOR and PKA upon nutrient shifts. I examine the evidence that these proteins are indeed direct targets of both pathways and which aspects of their regulation are targeted by TOR and PKA. I further explore if they are phosphorylated on shared sites by PKA and Sch9 or when experimental findings point towards regulation via the PP2A^Sit4^/PP2A branch downstream of TORC1. Finally, I critically review data suggesting direct cross-talk between the pathways and its potential mechanism.

## 1. Introduction

Protein kinase A (PKA) and TOR signaling are two highly conserved signaling pathways that respond to nutrient and stress signals and regulate various responses that govern cellular growth. Numerous findings indicate a strong connection between the pathways, but no clear picture of the nature of this interplay has emerged. This work aims to critically review the literature on shared targets and direct cross-talk and to point out gaps in current knowledge that hinder a better understanding. Beyond providing a resource about specifics of the signaling systems discussed, the described modes of interaction are intended to serve as examples relevant for understanding signaling interplay in a wider context. I will first provide a brief introduction to the PKA and TOR pathways to introduce the main players referred to subsequently. Then, I will give an overview of genetic data that link the pathways, before describing their major shared functions and substrates. Finally, I will discuss proposed direct cross-talk.

### 1.1. TOR Signaling

TOR signaling is one of the most central mechanisms that allows cells to adapt their growth to nutrient availability and also functions as a stress sensor. TOR signaling has been reviewed elsewhere [1,2,3,4,5,6,7,8,9,10,11] and therefore only a short summary is given here, in particular with respect to downstream functions that are shared with PKA signaling. The TOR functions explored in this review are mediated through TOR complex 1 (TORC1), and therefore “TOR signaling” will refer to signaling through TORC1 for the rest of this review. TORC1 exerts its physiological effects mainly through regulation of ribosome production, cell cycle progression and amino acid import and metabolism, as well as repression of autophagy and the cellular stress response [3,10]. 

Signaling downstream of TORC1 can be divided into two major branches, namely the PP2A and Sch9 branch. PP2A is a trimeric protein phosphatase, consisting of catalytic C subunit Pph21 or Pph22, scaffold A subunit Tpd3 and regulatory B subunit Cdc55 or Rts1 [12,13,14,15]. In addition, *S. cerevisiae* expresses a PP2A-like phosphatase (referred to as PP2A^Sit4^), consisting of catalytic subunit Sit4 and either Sap155, Sap185 or Sap190 [16,17]. Both PP2A and PP2A^Sit4^ activity are inhibited through Tap42 which forms complexes with Pph21/22 and Sit4 in a TORC1-dependent manner [18,19]. The *tap42-11* mutant, which renders Tap42 temperature sensitive, but also rapamycin insensitive, is a frequently used experimental tool in this context [18]. When TORC1 is inactivated, PP2A^Sit4^ induces a transcriptional program allowing the utilization of non-preferred nitrogen sources (among others through the transcription factors Gln3 and Gat1) and alters the profile of plasma membrane amino acid transporters [20,21,22,23]. 

The second major direct TORC1 target is the AGC kinase Sch9. Like other AGC kinases, it is basophilic and its limited number of known substrates suggest a preference for arginines and, to a lesser extent, lysines in the P-3 and P-2 positions [24,25]. It is phosphorylated by TORC1 on six serine and threonine residues near its C-terminus that reside within the so-called hydrophobic motif and turn motif [26]. In addition to the hydrophobic motif, AGC kinases generally require phosphorylation of their activation loop for full activity, which is catalyzed by the PDK1 homologs Pkh1/2 [27,28,29]. Sch9 is phylogenetically closely related to mammalian PKB/Akt and S6K [30] and, due to its ability to phosphorylate Rps6, is generally considered the functional homolog of S6K [26]. Several mechanisms through which Sch9 regulates ribosome biogenesis are discussed below.

### 1.2. PKA Signaling

PKA is a hetero-tetramer of two regulatory and two catalytic subunits. In *S. cerevisiae*, the regulatory subunit is encoded by *BCY1* and the catalytic subunits by *TPK1*, *TPK2* and *TPK3*, which are members of the AGC kinase family [31,32]. Specificity of PKA for phosphorylation of sites in R[RK]x[ST] (where x is any residue) motifs is well established [33,34]. Combinatorial deletions of the catalytic subunits demonstrated that only *tpk1∆ tpk2∆ tpk3∆* strains are inviable, while the viability of double deletion strains suggests a high level of redundancy [31]. 

PKA is activated by the binding of cAMP to the regulatory subunits, triggering their dissociation from the catalytic subunits [35]. The second messenger cAMP is produced by adenylate cyclase Cyr1, which is activated via two routes: First, by the small G proteins Ras1 or Ras2, which are regulated by the guanine-nucleotide exchange factor Cdc25 and GTPase-activating proteins Ira1/2, and second via the G protein-coupled receptor Gpr1 and its G protein alpha subunit Gpa2 [36,37] (Figure 1). Both pathways are best known for their activation by glucose when added to cultures without a fermentable carbon source [38,39,40]. The low-affinity, high-capacity phosphodiesterase Pde1 and a high-affinity, low-capacity phosphodiesterase Pde2 are responsible for cAMP degradation [41,42]. Through upregulation of Pde1 activity and other negative feedback loops, the PKA pathway dampens its activity within minutes after glucose addition, resulting in a characteristic cAMP spike [43]. While glucose-triggered activation is the by far most studied scenario, activation of PKA in a cAMP-independent manner [44] and in response to nitrogen and other nutrients [45,46] has also been described. There is an increasing number of examples in which conveying the presence of these nutrients to PKA depends on nutrient transceptors, transporters that serve a role in signaling (see [47] for a review). PKA is also phosphorylated by the PDK homologs Pkh1/2 and undergoes autophosphorylation, but our understanding of the regulatory roles of these modifications is limited [48,49].

Important tools for studying PKA are strains in which the pathway is artificially activated, through *BCY1* deletion or a single amino acid substitution in Ras1/2 (ras^V19^). These strains fail to grow on non-fermentable carbon sources and to accumulate storage carbohydrates, arrest in G0 or acquire heat-shock resistance like wild-type strains upon nutrient deprivation [32,50,51,52]. Growth of *bcy1∆* strains on non-fermentable carbon sources is restored by deletion of any two of the TPK genes and a point-mutation of the third, denoted as *tpk^w^* (“wimpy”). These mutants were isolated from spontaneous revertants of strains carrying deletions of *BCY1* and two TPK genes, which formed papillations after the parent strains had exhausted glucose on agar plates [53]. They form important tools for study as their remaining PKA activity can no longer be regulated by cAMP binding to Bcy1. Their capacity to accumulate glycogen upon nutrient exhaustion and utilize it upon nutrient repletion must therefore rely on signaling other than through PKA or on PKA regulation independent of cAMP [53]. 

Similar to TOR signaling, PKA has been implicated in the positive regulation of ribosome biogenesis and cell cycle progression and the repression of autophagy and the cellular stress response. It is also involved in pseudo-hyphal growth and meiosis. Further, PKA plays a major role in the regulation of metabolism; however, unlike for TOR signaling, this mainly evolves around the storage carbohydrates glycogen and trehalose, glycolysis and gluconeogenesis [1,11,54,55]. Several substrates regulated through direct phosphorylation by PKA have been identified [56,57,58,59]. 

Intriguingly, the inviability of the *tpk1∆ tpk2∆ tpk3∆* triple deletion strain can be rescued by the additional deletion of *YAK1*, *RIM15* or double deletion of *MSN2* and *MSN4* [60,61,62]. All these proteins, which are direct PKA substrates, play important roles in communicating stress signals and sending cells into quiescence, indicating that repression of these responses is the only essential PKA function [5,61,63,64,65]. 

Yak1, Msn2/4 and Rim15 are connected in a number of ways: while Yak1 phosphorylates and positively regulates Msn2/4 [66,67], Msn2/4 are conversely required for transcription of the *YAK1* gene [62]. Similarly, Rim15 also appears to phosphorylate Msn2/4 and Rim15-dependent regulation of gene expression is to a large extent explained by Msn2/4 [68,69]. 

The interconnectivity between the three factors may explain why the removal of any of them is sufficient to restore viability of the *tpk1∆ tpk2∆ tpk3∆* triple deletion strain. The fact that abolishing stress-induced anti-growth functions, is sufficient for viability of PKA-null strains, while its role in positively regulating growth is dispensable, prompts the question if another pathway supports growth in this context. TOR signaling is an obvious candidate and we will start exploring the relationship between the pathways by discussing the literature reporting their genetic interaction.

## 2. TOR–PKA Genetic Interactions

Findings about a genetic interaction between the TOR and PKA pathways pre-date even the discovery of TOR signaling itself, as overexpression of Sch9—later determined as the main TORC1 effector kinase—rescued a temperature-sensitive mutation of Cdc25 and deletions of components of the PKA pathway, including the catalytic subunits [70]. An overview of the many subsequently reported genetic interactions is given in Table 1. 

After the discovery of TOR signaling, a series of experiments in the mid-2000s using rapamycin further strengthened the connection between PKA and TOR signaling: Deletion of *BCY1* or overexpression of Cdc25, Tpk1 or an activated version of Ras (ras^V19^), all increased rapamycin resistance. These observations were most obvious in a *gat1Δ gln3Δ* background, indicating that PKA has the clearest effect on rescuing TOR inhibition when repression of the nitrogen discrimination pathway was rescued by independent means [71]. 

The *gat1Δ gln3Δ* mutations were, however, not strictly necessary, as deletion of *IRA2* or *BCY1* or the *ras2^V19^* mutation caused rapamycin resistance in an otherwise wt strain, while deletion of *RAS2* or of PKA catalytic subunits conferred rapamycin sensitivity [72]. 

Are these data consistent with a model in which PKA regulates TOR signaling or vice versa in a linear pathway? As the rescuing factor must act downstream or in parallel with the rescued factor, Sch9 should function downstream of PKA to rescue mutations in the PKA pathway, if assuming a linear connection [70]. In contrast, PKA should function downstream of TORC1 according to Schmelzle, 2004 [71] and Zurita-Martinez, 2005 [72]. The latter is not completely compelling, as temperature-sensitive or rapamycin-dependent inhibition may be incomplete and hyperactivation of an upstream function may alleviate a diminished downstream function. However, Toda, 1988 had also reported that hyperactivation of PKA via *BCY1* deletion rescued the growth defect of *sch9∆*-strains [70]. 

It is therefore clear that a linear connection between PKA and TOR signaling cannot explain the experimental observations, and instead, a parallel placement of the pathways may be assumed. Independent of the wiring, all of the above studies reported a positive interaction between TOR and PKA signaling. 

Conversely, antagonistic interactions have also been described: Araki et al. identified Pde2 and Bcy1 as suppressors of a temperature-sensitive mutation in the TORC1 subunit *KOG1* (aka *LAS24*) [73]. A later study found that genetic manipulations activating the PKA pathway (*bcy1∆* and expression of *ras^V19^*) increased rapamycin sensitivity, while *ras1Δ ras2-23* mutants and cells overexpressing *PDE2* were rapamycin resistant [74]. The latter also rescued the temperature sensitivity of a *tor2-ts* mutant, while the *ras^V19^* mutation caused synthetic growth defects with partial inhibition of TORC1. 

Therefore, the same genetic manipulations, *bcy1∆* and expression of *ras^V19^* from a single copy plasmid, lead to opposite outcomes in the studies by Zurita-Martinez et al. 2005 [72] and Ramachandran and Herman 2011 [74]: rapamycin resistance vs. rapamycin sensitivity. In addition to the use of different strain backgrounds, the major difference between the experiments was the use of different rapamycin concentrations, with at least ten times less in the latter study. It is interesting to note that this study observed increased phosphorylation of known PKA substrates upon rapamycin treatment, albeit on a timescale of hours [74]. As will be detailed below, there is in contrast ample evidence for reduced phosphorylation of substrates shared by PKA and TORC1/Sch9 upon rapamycin treatment. 

I propose a model in which the main interaction between TOR and PKA signaling is positive via shared substrates, but a second layer of weak mutual inhibition also exists. The latter may arise due to feedback from shared substrates. If one pathway is already deleted or strongly inhibited, further loss of input to the shared targets through inhibition of the second pathway will result in lethality or severe growth defects. In contrast, if, for example, TORC1 is only mildly inhibited (e.g., via low rapamycin), the activity of shared targets will be lowered, but sufficient to support growth when also PKA signaling is reduced (e.g., by *PDE2* overexpression). Negative feedback to TORC1 will be reduced, alleviating effects on TOR-unique targets and therefore resulting in rapamycin resistance. An analogous model may explain the observation that trehalase activity, generally considered a PKA-unique readout, was increased upon *SCH9* deletion [75]. Further work will be needed to test the proposed antagonistic/feedback effects. Signaling through shared functions and targets, which is, in contrast, more firmly established, will be discussed next.

## 3. Shared Targets

### 3.1. Ribosome Production

An increased rate of ribosome production to provide the machinery for growth is a hallmark of rapidly growing cells, compared to slowly growing cells under nutrient-limited conditions [76]. Ribosome production involves all three RNA polymerases for the synthesis of rRNA, ribosomal proteins and assembly factors. Considering that approximately half of all Pol II transcription initiation events take place at ribosomal protein gene promoters in rapidly growing cells and that rRNA makes up the majority of cellular RNA, it is unsurprising that these energy-intensive events are highly regulated in response to nutrient and other environmental conditions [77]. 

Consequently, ribosomal protein (RP) and ribosome biogenesis (RiBi; including rRNA modifiers, assembly factors and subunits of RNA polymerases I and III [78,79]) genes are two groups of genes most strongly affected by carbon source shifts. RP and RiBi genes each form regulons, i.e., groups of genes that appear highly coordinated in their expression [78,79]. Approximately 116 RP genes and >200 RiBi genes exhibit a rapid increase in expression after glucose addition to glucose-depleted cultures [79,80,81], while glucose-starved cells dramatically reduce the level of RP and RiBi transcripts within 30 min [79].

#### 3.1.1. Ribosome Biogenesis: Dot6/Tod6 and Stb3

Transcriptomic studies found that activation of the PKA pathway (e.g., via overexpression of activated Ras2 or activated Gpa2) recapitulated transcriptional changes induced by glucose addition to cultures without a fermentable carbon source for a large number of genes [81,82]. 

Conversely, the glucose-dependent transcriptional changes were largely blocked when simultaneously inhibiting PKA in one study. Despite its apparent genetic interaction with PKA, inhibition of Sch9 had little effect [82]. 

This observation was seemingly in stark contrast to a previous finding that a majority of genes induced by glucose addition in a wt strain were still induced in a *tpk^w^* strain in which cAMP-dependent regulation of PKA is disabled through *BCY1* deletion [81]. This group of genes was strongly enriched in RiBi genes. Given its genetic interaction with PKA, TOR/Sch9 signaling was an obvious suspect for the redundant pathway [81]. 

An interesting temporal perspective on the contribution of PKA and TOR signaling to RiBi transcriptional regulation has recently been unveiled: after an initial phase upon glucose addition, in which PKA was the dominant factor necessary for RiBi gene transcription, a co-operative effect between the pathways, with TOR signaling gaining importance, was observed [83]. As inhibition of PKA and Sch9 by Zaman et al. was performed 20 min after glucose addition, much effect of PKA and little of Sch9 inhibition was observed since the PKA contribution was dominant at this timepoint [82]. 

At face value, this does not resolve the discrepancy to the study by Wang et al., as both early and late induction of RiBi genes in *tpk^w^* were observed here [81]. Invoking the earlier proposal of indirect negative TOR–PKA interaction, it can, however, not be ruled out that the temporal dynamics of TOR signaling in response to glucose are altered in this strain in adaptation to the mutations in the PKA pathway. These effects would be less obvious in the context of instantaneous inhibition used by Zaman, 2009 [82]. 

If the temporal observations by Kunkel et al. hold true for other transcript classes remains to be determined. Interestingly, as detailed in the following, the dynamics observed for RiBi genes are not adequately explained by current knowledge of their PKA- and TOR-dependent mechanisms of transcription regulation [83].

The architectures of Ribi gene promoters are distinct from the one of RP genes, with Rap1-binding sites only present in a small subset of RiBi-promoters [84]. Instead, RiBi-promoters are enriched in PAC (Polymerase A and C) and RRPE (rRNA processing element) motifs [80,81,85,86,87]. The PAC and RRPE elements are each found in approximately half of the RiBi-promoters and approximately one-quarter of the promoters contain both elements [79]. Both elements function in the binding of transcriptional repressors, Dot6 and its homolog Tod6 in the former, and Stb3 in the latter case [24,88]. These transcription factors exert their repressive role via recruitment of the Rpd3L histone deacetylase complex [24]. This is in agreement with observations that Rpd3L is recruited to a number of rapamycin-repressed genes upon rapamycin treatment [89] but in contradiction to earlier studies that found Rpd3L binding to be constitutive [90,91]. 

Double deletion of *DOT6* and *TOD6* led to a noticeable change in transcriptional repression after glucose and nitrogen starvation [92] and both proteins were dephosphorylated as early as 5 min after these starvations and upon various stresses, such as heat, oxidative and osmotic stress, as well as upon rapamycin treatment [93]. The same study demonstrated nuclear localization of Dot6 and Tod6 following stress, nitrogen- or glucose starvation. 

It was shown recently that the levels of Dot6 and Tod6 are low when cells are grown in medium without a fermentable carbon source and only rise upon glucose addition and that early RiBi gene induction upon glucose addition occurred normally in a *dot6∆ tod6∆* strain [83]. Therefore, the regulatory function of these transcription factors is likely relevant upon depletion of a fermentable carbon source rather than in the relief of repression upon encountering glucose. The mechanism of RiBi gene regulation in the latter transition is still unclear [83]. 

One of the early phospho-proteomics screens into TOR signaling found reduced phosphorylation of Dot6 upon rapamycin treatment, which was alleviated by mutations of Sch9 that rendered it active independent of TORC1 [94]. Similar observations were made for its paralog Tod6, but here hypo-phosphorylation was additionally alleviated by rapamycin-insensitive tap42-11. These observations were only partially reflected by gel-shift assays of Dot6, while Tod6 exhibited clear dephosphorylation, that depended on the inhibition of Sch9. Dot6 was also detected in a screen for proteins interacting with a substrate-trapping mutant of Tpk1 and subsequently shown to be phosphorylated by PKA in vitro [95]. Later, in vitro phosphorylation of Dot6/Tod6 by Sch9, as well as a decrease in Dot6/Tod6 phosphorylation after inhibition of Sch9 or PKA in vivo was also observed [24]. The consequence of *DOT6* and/or *TOD6* deletion for RiBi gene repression upon rapamycin treatment or analog-sensitive PKA inhibition have been evaluated, and a more prominent role was attributed to Tod6 downstream of TORC1 and to Dot6 downstream of PKA [92]. 

A further transcriptional repressor, Stb3, that forms part of the RPD3L complex [96] was also found to be phosphorylated by PKA and Sch9 in vitro [24,97]. Stb3 phosphorylation also decreased upon Sch9 inhibition in vivo and Stb3 was recruited to both RiBi and RP-promoters upon Sch9 inhibition, which correlated with the recruitment of RPD3L [24]. A growth defect caused by Stb3 overexpression was mitigated by the deletion of *PPH22*, but more direct involvement of PP2A in the regulation of Stb3 is lacking [98]. Stb3 is regulated via subcellular localization as glucose addition to post-log phase cells triggered its export from the nucleus within minutes, while rapamycin addition to log-phase cells had the opposite effect [98] (Figure 2). 

The phosphorylation sites on Dot6, Tod6 and Stb3, which are presumably directly phosphorylated by PKA and Sch9 (and potentially dephosphorylated by PP2A), have not yet been accurately mapped. Dot6 and Tod6 are among the proteins with the highest number of PKA motifs in yeast. Mutation of six phosphorylation sites in R[R/K]x[S/T] motifs on Tod6 and four sites in R[R/K]x[S/T] and one in an Rxx[S/T] motif on Dot6 caused a decrease, but not complete abrogation of phosphorylation by Sch9 in vitro [24]. The identity of sites phosphorylated by PKA on Dot6 and Tod6 has not been reported to date. The RRxS motif at S283 in Stb3 is conserved in yeast species from *S. cerevisiae* to *C. albicans* and was therefore proposed as a functionally important target in PKA signaling [97]. Further phospho sites with PKA consensus motifs exist on Stb3 but show a lower degree of evolutionary conservation. Signal obtained from an antibody directed against the RRxS-motif on purified Stb3 was completely lost after Sch9 inhibition, indicating that Sch9 targets the same motif on this protein. Surprisingly, inhibition of PKA alone did not cause a clear change in signal [24]. Further, the addition of cAMP to a *cyr1∆* strain was not sufficient to reverse Stb3 nuclear localization under conditions of glucose depletion or rapamycin treatment [98]. In contrast, glucose re-addition to the same glucose-depleted strain (without cAMP) triggered Stb3 cytoplasmic localization [98]. Therefore, cAMP neither is necessary nor sufficient for Stb3 translocation. Together, these data suggest that TOR signaling overrides PKA-dependent regulation of Stb3 under the conditions tested.

#### 3.1.2. Ribosomal Protein Production: Ifh1, Crf1 and Spf1

The second major regulon in ribosome production are transcripts coding for ribosomal proteins (RP). As for RiBi genes, rapamycin treatment causes severe repression of this regulon [21,99]. The main transcription factors involved in RP gene transcription are Rap1, Hmo1 [100], Sfp1 and Fhl1 with its co-factors Ifh1 and Crf1 [101,102,103]. Of these, Rap1, Sfp1 and Fhl1-Ifh1/Crf1 have been linked to the TOR and PKA pathways [101,102,104,105,106]. Additionally, Stb3, functions as a transcriptional repressor also of RP genes, albeit apparently not via the binding of RRPE promoter motifs [24,88] (Figure 2). PKA-dependent regulation of RP genes induced by Rap1 has been shown in a number of experiments [106,107]: RP-transcripts were up-regulated in a Rap1-dependent manner in a *BCY1* deletion strain and the presence of the Rap1 binding site in its promoter was sufficient to confer an increase in a reporter transcript in a *bcy1∆* strain compared to wt [108]. While phosphorylation of Rap1 via PKA cannot be completely ruled out, more recent models suggest, Rap1 serves as a recruiting factor for proteins regulated by PKA. These binding partners may include Sfp1 and Fhl1 and its co-factors [107].

While 127 of the 138 RP genes harbor Rap1-binding sites, approximately a half are bound by Fhl1 [101,107]. Fhl1 is not apparently regulated by PKA or TOR, instead, regulation occurs on the level of its co-regulators Ifh1 and Crf1. The level of the Ifh1 bound to RP genes drops upon nutrient depletion or rapamycin treatment and increases when cells resume growth upon encountering improved nutrient conditions, while Fhl1 remains constitutively associated with the promoters [101,102,103]. The interaction of Fhl1 and Ifh1 depends on the forkhead-associated (FHA) domain of Fhl1 [101,102], a domain previously reported to interact with phosphorylated sequences [109]. It is, therefore, reasonable to posit that Fhl1-Ifh1 interaction depends on Ifh1 phosphorylation. Martin, 2004 provided a detailed description of how transcriptional activator Ifh1 and repressor Crf1 compete for binding to Fhl1 [104]. They observed a rapamycin-triggered increase in Crf1 phosphorylation, causing its nuclear translocation and therefore RP gene repression. As this process could be prevented by mutations that render PKA signaling hyperactive, they suggested that TORC1 acts upstream of PKA in this respect [104]. They further found that protein kinase Yak1 is required for Crf1 nuclear translocation. As Yak1 is a known direct PKA, but not TORC1/Sch9 target, this is also consistent with the idea of TORC1 regulating PKA. 

Surprisingly, regulation of RP gene transcription by Crf1 appears to be yeast strain dependent, as Zhao et al. observed no effect of *CRF1* deletion on RP transcription in the W303, while they did observe the effect in the TB50 strain [110]. The Yak1-dependent mechanism also appears to be at odds with the finding by others that RP gene transcription is still repressed by rapamycin in strains lacking Yak1 [72]. I propose that this is explained by a secondary mechanism that acts on longer timescales. Indeed, remaining RP gene inhibition in strains lacking *CRF1* was observed also in the original study by Martin, 2004. What this second mechanism may be is not yet clear. Finally, to date, only CK2 has been shown to phosphorylate Ifh1 and Crf1 in a manner relevant for Fhl1 binding [111,112]. Mechanistic details if and how PKA and TORC1 regulate Fhl1-dependent RP transcription are therefore still lacking.

Sfp1 is a transcription activator of RP genes and deletion strains of SFP1 are marked by a striking small cell size phenotype [113]. While Sfp1 binds promoters of other gene groups as well, this appears to be mechanistically different [114]. 

Sfp1 was associated with TOR signaling in a microscopic screen as it localized from the nucleus to the cytoplasm upon rapamycin treatment [115]. Importantly, PKA signaling can override TORC1 effects to some extent as hyperactivated PKA (*bcy1∆*) largely counteracted the rapamycin-induced cytoplasmic relocalization. In which way do PKA and TOR signaling converge on Sfp1? Sfp1 phosphorylation was reduced upon rapamycin treatment and, in vitro phosphorylation by TORC1 was observed, while no phosphorylation was detected in an in vitro kinase assay with Sch9 [105]. This renders Sfp1 one of only a handful of direct TORC1 substrates. Seven potential TORC1 phosphorylation sites on Sfp1 were explored and their mutation almost completely abrogated further dephosphorylation upon rapamycin treatment [105]. Sfp1 was also phosphorylated by bovine PKA in vitro, but phosphorylation sites were not determined [97]. An RRxS motif at a site different from the TORC1-dependent sites and conserved in multiple yeast species was proposed to constitute an important PKA site [97]. Regulation of Sfp1 through common sites by PKA and TORC1 is therefore unlikely. 

The above data on Sfp1 localization may also be explained by regulation of PKA by TORC1. It was observed, that in the absence of PKA (*tpk1∆ tpk2∆ tpk3∆ msn2∆msn4∆*), Sfp1 was more cytoplasmic than in wt cells; however, rapamycin caused further cytoplasmic localization [115]. Therefore, the TORC1–PKA relationship on Sfp1 is at least not purely epistatic. 

Two major conundrums about the TORC1-dependent regulation of Sfp1 remain: First, while mutation of the TORC1-dependent phospho sites on Sfp1 rendered the protein constitutively cytoplasmic, the corresponding strain did not exhibit the striking small cell size phenotype of sfp1∆ cells [105]. Second, while rapamycin treatment caused loss of Sfp1 phosphorylation, glucose or complete nutrient starvation had no obvious effect on Sfp1 phosphorylation. Nonetheless, nutrient starvation was associated with Sfp1 relocalization [105].

#### 3.1.3. RNA Pol III Transcription: Maf1

TOR and PKA signaling are also implicated in the regulation of RNA polymerase III (Pol III), mainly through its highly conserved negative regulator Maf1. Maf1 is required for transcription repression under a variety of stress conditions, including rapamycin treatment [116,117]. 

Its function is also repressed by PKA through direct phosphorylation near one of its two nuclear localization signals (NLS), counteracting Maf1 nuclear accumulation [118,119]. As mutants in the alleged phospho sites accumulate in the nucleus without repressing RNA Pol III, an additional layer of Maf1 regulation is believed to exist [118]. Maf1 is also phosphorylated by Sch9 in vivo and in vitro, presumably on sites that are identical to or at least strongly overlap the PKA sites and reside within PKA consensus motifs [94,120]. Phospho-mimicking mutation of the seven potential phospho-acceptor sites prevented the interaction of Maf1 with Pol III subunit Rpc82 [94]. 

If both pathways are responsible for the phosphorylation of Maf1 on the same sites, how can specific inhibition of only TORC1 by rapamycin nonetheless have an obvious effect? Part of the answer may lie in the phosphatase branch, which exists in addition to the Sch9 branch downstream of TORC1. A Maf1-dependent reduction in RNA Pol III transcription by *SCH9* deletion has been reported; however, rapamycin treatment exhibited an additional effect in these strains. This led to the notion of TORC1 regulating Maf1 both in an Sch9-dependent and -independent manner [120]. Therefore, activation of the phosphatase branch may be instrumental in removing remaining, PKA-dependent phosphorylation. 

Even though the tap42-11 mutation, which renders Tap42 rapamycin insensitive, did not affect Maf1 dephosphorylation [94], mutations of protein phosphatase PP2A catalytic subunits prevented Maf1 dephosphorylation upon rapamycin treatment. They not only blocked Maf1 inactivation but also its nuclear localization, suggesting that PP2A counteracts phosphorylation by PKA and Sch9 [121] (Figure 3). Constitutive activation of PKA via a *BCY1* deletion was sufficient to prevent a reduction in Pol III transcription upon rapamycin treatment [118], indicating that highly active PKA can override PP2A action. A prediction of the resulting model is that inactivation of either Sch9 or PKA alone without PP2A activity should be insufficient for Maf1 dephosphorylation. This remains to be adequately addressed, as there is still disagreement about the exact contribution of each kinase [94,120]. Either way, this example demonstrates that the phosphatase branch needs to be considered with respect to TORC1–PKA interaction, despite apparent convergence of the pathways via Sch9 and PKA otherwise.

### 3.2. Autophagy

While the functions described above exemplify roles of TOR and PKA in promoting anabolism, both pathways also repress catabolism, most notably through inhibition of macro-autophagy (in the following simply referred to as autophagy). Conversely, rapamycin treatment and nitrogen starvation are potent inducers of autophagy [122]. No induction of autophagy was observed upon activation of the PP2A branch downstream of TORC1, indicating that this function is controlled by a different or additional downstream branch or directly by TORC1 [123].

Hyperactivation of the PKA pathway led to rapamycin resistance with respect to autophagy induction [71], blocking autophagy at a step upstream of autophagosome formation [124,125]. Inhibition of PKA under some conditions was [124,126] and others, possibly due to incomplete inhibition, was not [125] sufficient to induce autophagy. In the latter case, simultaneous inhibition of PKA and Sch9, resulted in visible levels of autophagy, albeit less than caused by rapamycin treatment [125].

The interaction of Atg1 and Atg13 is an early step in the cascade of autophagy induction and is required for Atg1 kinase activity [123,127]. Atg13 also acts as a bridge to Atg17 in the Atg17-Atg29-Atg31 complex in the assembly of the pre-autophagosomal structure (PAS) [128,129]. 

The in vivo phosphorylation of Atg1 and Atg13 are highly dependent on nutrient availability, implicating them as candidates for TOR- and PKA-dependent phosphorylation in a step far upstream in the autophagic cascade [97,123].

Atg1 was found to be phosphorylated by PKA in vitro and in vivo [97]. While the kinase activity of Atg1 was not apparently affected by mutation of the PKA sites S508 and S515, the same alanine-mutations allowed PAS-formation even in the context of hyperactive PKA [97,130]. Therefore, the phosphorylation of these sites by PKA is likely a mechanism to counteract autophagy. In addition, four distinct rapamycin-sensitive phospho sites, S474, S518, S677 and S680 were identified on Atg1 and TORC1-dependent phosphorylation is conserved in its mammalian homolog [131].

Atg13 is dephosphorylated within minutes of nitrogen starvation or rapamycin treatment and concomitantly, Atg1–Atg13 and Atg13–Atg17 interactions are triggered [123,129]. Phospho-mimicking mutations in the Atg1- and Atg17-interacting regions of Atg13 diminished the respective interactions, strongly suggesting that TORC1-dependent phosphorylation counteracts these binding events required for PAS assembly [129]. Phospho-null mutation of the respective serines did, however, not lead to constitutive Atg1–Atg13 interaction or autophagy induction [129], possibly because of the existence of additional phospho sites [132].

Similarly, Atg13 also harbors conserved PKA consensus sites and is phosphorylated by PKA in vitro and in vivo, with the substrate sites mapped to residues distinct from those reported as TORC1 targets (Figure 4) [97,126,129,132]. Inhibition of PKA led to an increase in Atg1 autophosphorylation and this may be a consequence of altered Atg13 phosphorylation [126]. Mutation of PKA sites on Atg13 to alanine caused constitutive association of Atg13 with the PAS, but no observable induction of the interaction with Atg1 [126].

With regulation of Atg13 via phosphorylation by TORC1 and PKA therefore firmly established, what is the relationship of the two pathways converging on this target? The Herman lab observed a synergistic effect of TORC1 and PKA, with inhibition of both pathways leading to more pronounced autophagy induction than either one on its own [126]. In addition, rapamycin treatment did not alter Atg13 phosphorylation detected with an anti-PKA-substrate antibody. Conversely, unlike rapamycin treatment, PKA inhibition did not affect the electrophoretic mobility of Atg13 [126]. These findings strengthen the conclusion that TORC1 and PKA likely target distinct sites on Atg13. The reported TORC1-dependent sites fall both within the interaction regions with Atg1 and Atg17, while one PKA site is located at the edge of the Atg17 interaction region and the remaining in uncharacterized parts of the Atg13 intrinsically disordered region (Figure 4) [126,129]. It may therefore be speculated that TORC1 regulates both the Atg1–Atg13 and Atg13–Atg17 interaction, while PKA only impacts PAS formation via the Atg13–Atg17 interaction. An important question that remains to be explored is whether the distinction of autophagy induction with inactivation of both TORC1 and PKA vs. only one of the pathways is physiologically relevant. 

Additionally, rapamycin-sensitive phospho sites were also detected on Atg2, Atg9 and Atg29 and 20 Atg proteins were found to be phosphorylated by TORC1 in vitro. Functional relevance in autophagy was demonstrated for one of the TORC1-dependent sites on Atg29 [133].

In summary, the PKA and TORC1 pathways are important negative regulators of macroautophagy. Despite much progress on their impact on Atg13, mechanistic details of other aspects in which they may regulate autophagy are still unresolved. In addition to macroautophagy, TOR signaling has also been implicated in other forms of autophagy, such as microautophagy and selective autophagy, while little is known about the relation of PKA with these processes. For further information on these subjects, I refer the reader to a recent review on selective autophagy [134] and a previous review discussing the role of TORC1 in different forms of autophagy [10]. 

### 3.3. Stress Response

A final shared function of TOR and PKA signaling is repression of the stress response and quiescence entry. Lethality of the *tpk1∆ tpk2∆ tpk3∆* triple deletion is rescued by the additional deletion of *YAK1*, *RIM15* or double deletion of *MSN2* and *MSN4* [60,61,62], rendering their repression the only essential function of PKA. In turn, Sch9 overexpression also restores viability of the PKA-null strain, indicating that the pathways also converge on this process [70].

#### 3.3.1. Rim15

Greatwall-kinase homolog, Rim15 was originally identified via its impact on meiosis in diploid cells [135]. Later it became apparent that Rim15 additionally serves as a master regulator of the entry to quiescence (G0-phase) [61]. *RIM15* deletion strains, such as strains with hyperactivated PKA, are defective in the proper establishment of the G0-program and its associated stress resistance upon growth into stationary phase, which is reflected, e.g., by reduced accumulation of trehalose, glycogen and stress response transcripts (e.g., *SSA3, HSP12* and *HSP26*) [61]. The induction of stress resistance through the Rim15-dependent transcriptional program also includes upregulation of superoxide dismutases Sod1 and Sod2 [69]. In the context of the diauxic shift, this represents an important adaptation to the increased level of reactive oxygen species due to the switch to respiratory growth [136]. Considering the reduced life span of *rim15∆* cells, the stress-protective role of Rim15 may be central to the life-span extending effects of mutations in the TOR and PKA pathways [69]. The major mechanisms through which Rim15 induces G0-arrest are the transcription factors Msn2/4 and Gis1, regulating stress response element (STRE) and post-diauxic shift (PDS) element genes, respectively [68,69,137]. Further, endosulfines Igo1/2 are direct Rim15 targets and in turn, regulate RNA processing and cell cycle progression [138,139,140,141] (Figure 5). For a detailed description of Rim15 downstream functions, I refer to a previous review [5].

Rim15 is inactivated by PKA and TORC1/Sch9 and it may therefore be expected that when only one of the pathways is turned off, Rim15 should remain inactive due to the activity of the other. Conversely, inactivation of either PKA or TOR signaling is sufficient to induce G0-arrest [35,142].

Interestingly, Rim15 is repressed by PKA and TORC1/Sch9 via apparently distinct mechanisms: Its catalytic activity is inactivated by phosphorylation of five residues (S709, S1094, S1416, S1463, S1621; all being part of PKA consensus motifs) by PKA [61], while its cytoplasmic retention is promoted by phosphorylation of T1075 downstream of TORC1 and direct phosphorylation of S1061 by Sch9 [137,143,144]. (Rim15 T1075 is also phosphorylated by Pho80/85, a cyclin-dependent kinase inactivated upon phosphate starvation [145,146,147]).

Rim15 is retained in the cytoplasm by 14-3-3 proteins Bmh1/2 binding to phosphorylated S1061 and T1075 [5,146]. Additionally, auto-phosphorylation of Rim15 has also been proposed to lead to nuclear export [147] (Figure 5).

Off note, Rim15 contains a PAS domain which may by itself act as a stress sensor [69,148]. It is therefore likely that Rim15 is activated also in the context of stresses other than nutrient limitation.

A number of options for activation of Rim15 upon inactivation of only TOR or PKA signaling may be proposed. First, one of the pathways may not be highly active at the time of inactivation of the other. For example, assuming a traditional model in which carbon source-dependent PKA activity mainly follows the profile of cAMP concentration, this activity should be highest immediately after glucose addition to cells grown without a fermentable carbon source and considerably lower during steady state growth [55]. Low-level PKA activity may not be sufficient to keep Rim15 inactive on its own. As discussed below, TORC1/Sch9 may be necessary to sustain the even limited PKA activity in the absence of a cAMP-pulse. 

Another explanation may be that the pathways do not impinge on Rim15 independently and instead, it is tempting to propose co-operation of TOR and PKA signaling on the level of Rim15: as nuclear PKA is believed to be held inactive by its regulatory subunit Bcy1, Rim15 is possibly only inactivated by the smaller cytoplasmic fraction of PKA catalytic subunits [149,150]. Rim15 cytoplasmic retention via TORC1/Sch9/Pho80/85 may therefore be required to extend the cytoplasmic resident time of Rim15 for sufficient phosphorylation by PKA. 

On the other hand, activation of Rim15 by inhibition of PKA alone may simply be explained by proposing that Rim15 remains active despite cytoplasmic retention, provided it is not phosphorylated by PKA. Apparently, the regulation of PKA activity via its subcellular localization is crucial in this model. Possible modulation of PKA subunit localization by TOR signaling will be discussed below.

If Rim15 is indeed activated upon inhibition of either PKA or TOR signaling alone, as well as which of the above speculations may provide a mechanistic explanation, requires answering technically challenging questions, such as “Is Rim15 active despite being bound by Bmh1/2?” and “Is PKA only active in the cytoplasm and is Rim15 cytoplasmic localization required for sufficient PKA-dependent phosphorylation?”.

Either way, Rim15 represents an intriguing PKA and TORC1/Sch9 target, in that apparently different sites on the same protein are exclusively phosphorylated by one of the kinases. How PKA and Sch9 phosphorylate distinct sites on Rim15, despite their shared consensus motifs remains an open question. It should be pointed out in this context, that while the phosphorylation of the mentioned sites by the indicated kinases, has been demonstrated, experiments addressing the absence of phosphorylation by the respective other kinases are largely missing [61,143,147].

#### 3.3.2. Msn2/4

Msn2 and Msn4 are homologous, partially redundant zinc finger transcription factors that mediate the response to various stresses such as heat, hyperosmotic stress or glucose starvation [151,152]. In the process, they localize to the nucleus in a rapid and rapidly reversible manner (<5 min) [64]. Msn2/4 regulate the transcription of over 150 genes and bind to a motif called the stress response element (STRE) in the promoters of many genes, such as *CTT1*, *HSP26* and *SSA3* [62,78,153,154].

As mentioned above, double deletion of *MSN2* and *MSN4* rescues the lethality of a *tpk1∆ tpk2∆ tpk3∆* strain [62]. Msn2 was found to be constitutively nuclear in a strain with impaired PKA activity, indicating that PKA counteracts Msn2 nuclear localization. Conversely, exogenous addition of cAMP to a strain lacking *PDE2* was sufficient to prevent and reverse Msn2/4 nuclear localization [64]. Rapamycin treatment also causes nuclear localization [22].

The molecular events governing Msn2 localization were investigated using two artificial GFP-fusion constructs. A construct of the Msn2 NLS (residues 567-648, which include PKA motifs at S582, S620, S625 and S633) and zinc finger domain coupled to GFP exhibited glucose-dependent localization like the full-length protein but failed to respond to other stresses or rapamycin treatment [64,155].

A second GFP-construct of the SV40 NLS and Msn2-NES localized to the cytoplasm in unstressed growth in glucose, but nuclear localization was triggered by glucose starvation, heat- or sorbate stress and rapamycin treatment. Interestingly, cAMP addition to a pde2∆-strain could override stress-induced nuclear localization [155]. The nuclear export of this construct is likely regulated via phosphorylation of the RRxS site S288 near the NES [64].

These experiments demonstrate that localization via both the NLS and NES is regulated by PKA. In contrast, the insensitivity of the localization of the NLS-construct to rapamycin and the rapamycin-dependent re-localization of the NES-construct indicate that only the latter is regulated by TORC1 [155] (Figure 6).

The role of TOR signaling in Msn2 localization remains controversial: An early study reported that rapamycin treatment of unstressed cells led to the nuclear localization of Msn2 [22]. Other studies failed to observe Msn2 localization to the nucleus upon rapamycin treatment alone, instead, rapamycin increased the propensity of re-localization upon stress [156,157]. While this discrepancy can possibly be explained by different timepoints of imaging, the studies also disagree on the role of the Tap42 branch in the potential transmission of signals from TORC1 to Msn2/4: Beck et al. found that Msn2 nuclear localization was unaffected by a *TAP42* mutation that renders it rapamycin insensitive (*tap42-11*) or a *sit4* mutation [22]. Instead, in the second study, a temperature-sensitive mutation in *TAP42* or rapamycin treatment impeded the return of Msn2/4-dependent transcripts to baseline levels and of Msn2 to the cytoplasm, after a heat shock [156]. This suggests inefficient re-phosphorylation, as a phosphatase otherwise inhibited by Tap42 (i.e., PP2A or PP2A ^Sit4^) remains active when TORC1 is inhibited (Figure 6).

Data from the Hall and Broach labs are in agreement that SIT4-mutation does not prevent re-localization [22,157]. Instead, a *pph21 pph22* double deletion as well as *tpd3* and *cdc55* single deletions reduced stress-induced nuclear localization [157,158]. Deletion of the other PP2A regulatory subunit, Rts1, did not show any effect. This strongly suggests that PP2A^Cdc55^ is the relevant phosphatase for dephosphorylating the Msn2/4 NES.

However, it was noted that Msn2-GFP was able to accumulate in the nucleus in the context of glucose starvation in *pph21∆ pph22∆* cells, presumably because the glucose-dependent regulation of the NLS is independent of this phosphatase [157]. A number of points still need to be clarified with respect to a model in which PP2A^Cdc55^ regulates Msn2/4 via the NES: First, a strong effect of PP2A^Cdc55^ on the phosphorylation state of the Msn2/4 NES remains to be demonstrated. Additionally, a *CDC55* deletion strain retained its effect on Msn2/4-dependent transcription upon deletion of the NES. As PP2A^Cdc55^ regulates chromatin association of Msn2/4, it has been speculated that PP2A^Cdc55^ is also involved in dephosphorylating the Msn2/4 zinc-finger domain [158].

Given the regulation of PP2A by TORC1 and the nitrogen-sensing function of TORC1, it is unsurprising that nitrogen starvation also caused Msn2 nuclear localization [156]. It remains, however, an open question why inhibition of TORC1 by nitrogen starvation was sufficient to cause this effect in this study, while inhibition of TORC1 by rapamycin was not. The answer to this question may be linked to why rapamycin alone was sufficient in one [22], but not other studies [156,157]. 

Further levels of post-translational regulation of Msn2/4 beyond PKA and PP2A exist via PP1, which dephosphorylates the NLS and Snf1, which phosphorylates S582 on Msn2 [159] (Figure 6). As both pathways are carbon-source regulated, they complicate the interpretation of the role of PKA and TOR signaling in response to carbon source shifts. Finally, direct phosphorylation by Rim15 and Yak1 have also been demonstrated [66,68].

### 3.4. Summary of Shared PKA and TOR Targets

In the previous sections, I introduced shared targets of the PKA and TOR pathways and how in many cases loss of only one of the pathways causes loss of the function imparted by PKA and TOR on the target. This may be achieved via a finely tuned additive effect, but is unlikely as this wiring would not be robust to, e.g., fluctuation in kinase levels. Instead, there are hints at other—not simply additive—convergence mechanisms. Similar mechanisms are likely to exist in the context of different scenarios in which signaling pathways converge on shared cellular functions. Examples are summarized in Figure 7 and substrates for which each mechanism may be most relevant have been tentatively assigned. As outlaid in the corresponding sections, much further work is needed to back or disprove these models.

Some substrates (e.g., Dot6/Tod6) may be phosphorylated by the two pathways at different times after a stimulus. This idea can be extended to an override-model in which highly stimulated PKA masks TOR-inactivity, which may be relevant in many cases of artificial PKA activation. Alternatively, or additionally, phosphorylation by the two pathways may take place in different cellular locations. Rim15, Msn2/4 and Atg13 may also be phosphorylated on distinct sites by the two pathways. Further, as has become clear in the previous sections, it is the Sch9 branch, rather than the complete TOR pathway that most immediately converges with PKA. As is the case for Maf1, the PP2A branch may therefore rather be viewed as modulating the PKA-Sch9 interplay. Finally, it may be necessary to consider additional inputs to the pathways. In particular, activation-loop phosphorylation of Sch9 and PKA by Pkh1/2 [28,48,49] and regulation of Sch9 by Snf1 have been reported [160].

## 4. Substrate Specificity of PKA and Sch9

Up to now, I discussed how PKA and TOR signaling converge on shared substrates. It is, however, similarly important to ask how specificity is achieved for substrates that are not shared. Two phospho-proteomics studies obtained clear indications that the prevalence of shared substrates is extensive, based on the finding that the PKA motif R[R/K]x[S/T] [161,162] was enriched among sites hypo-phosphorylated upon rapamycin treatment [25,133]. Importantly, however, not all known PKA targets were affected by rapamycin treatment, which was validated for a subset [25]. Similarly, we found that some, but not all sites hypo-phosphorylated upon PKA inhibition were also hypo-phosphorylated upon Sch9 inhibition [163]. No *bona fide* unique Sch9 site is known to date and the sites shared with PKA reside in the R[R/K]x[S/T] motif [94,120]. (In contrast, proteins downstream of TORC1, but not PKA (e.g., Npr1, Gat1, Gln3, Rtg1 [18,22,164]), are connected to TORC1 is via PP2A/PP2A^Sit4^ rather than Sch9.) The question, therefore, becomes, how PKA can achieve specificity for other sites with the same motif, such as in Pfk26 [56], Nth1 [58,165], Cki1 [166], Adr1 [167] and Ssn2/Srb9 [168]. If differences in substrate motifs of PKA and Sch9 do not explain why some targets are exclusively phosphorylated by PKA, different localization of the kinases relative to their targets may be invoked. Sch9 is present both in association with the vacuolar membrane and dispersed throughout the cytoplasm [26,169,170] and a recent study detected pools of Sch9 at additional locations, including the plasma membrane and nucleo-vacuolar junction [171]. As discussed below, our understanding of the subcellular localizations of PKA is still far from complete, but it is at least partially found in the cytoplasm. As also several substrates unique to PKA (e.g., Nth1, Pfk26), mainly localize to the cytoplasm [170], localization of the kinases does not appear to be the explanation for specificity, unless we were to assume that they are only active at a subset of locations. For example, one might propose that Sch9 is mainly active at the vacuolar membrane, where TORC1 resides, and PKA mainly in the vicinity of Cyr1.

## 5. Potential Mechanisms of TOR–PKA Interplay

Having explored how PKA and TOR signaling interact via shared targets, we will now return to discussing a potential more direct cross-talk between the pathways. As pointed out in an earlier section, the regulation of one pathway by the other is not sufficient to explain experimental observations. However, this does not rule out that such cross-talk may exist in addition to convergence on shared substrates. TOR was initially discovered by the Hall lab via a genetic screen for spontaneous rapamycin-insensitive mutants, leading to the identification of the target-of-rapamycin kinases Tor1 and Tor2 [172]. In subsequent years, knowledge about TOR signaling and its response to the availability of nitrogen sources grew and its suppression of the transcription factors Gln3 and Gat1 was described [22]. In a later study, the same lab asked if further rapamycin-insensitive mutants could be detected in a *gln3Δ gat1Δ* background and several genetic manipulations hyperactivating PKA signaling (*ras^V19^* mutation, *CDC25* or *TPK1* overexpression) were found to further increase rapamycin resistance [71]. Based on this, they proposed a model in which TORC1 acts as an upstream activator of PKA. While none of their data could distinguish between this model and convergence of TOR and PKA signaling on shared substrates, they observed that rapamycin treatment caused Tpk1 nuclear localization, and therefore, a mechanism by which TOR may regulate PKA. This rapamycin-induced nuclear localization was reproduced later, with the additional observation that *SCH9* deletion resembles the effect of rapamycin treatment in this respect [25]. It was also noted that mutations that abolish Tpk1-Bcy1 interaction prevented rapamycin-induced Tpk1 nuclear localization. This suggests that nuclear Tpk1 is bound and inhibited by Bcy1 [25].

The model of TORC1 regulating PKA gained support through a subsequent study: Martin et al. showed that hyperactivation of PKA prohibited rapamycin-induced nuclear translocation of the RP gene repressor Crf1 and that PKA inactivation led to nuclear translocation without the need for rapamycin [104] (see above). These data rule out a connection of PKA and TORC1 to Crf1 via a strict AND or strict OR gate, but may be explained by a more elaborate mechanism, such as shown in Figure 7 on a negative regulator of Crf1. Importantly, however, the same work further found that phosphorylation by Yak1 appears responsible for Crf1 translocation and that Yak1 is activated in presence of rapamycin. This argues for TORC1 acting at the level or upstream of Yak1. Notably, Schmelzle et al. had observed that *YAK1* deletion is equivalent to PKA hyperactivation with respect to overcoming Gat1-Gln3-independent rapamycin effects [71]. The fact that Yak1 has been described to be inactivated by direct phosphorylation by PKA [63,65,173], but not been reported as a TORC1 or Sch9 target, supports TORC1 dependent regulation of PKA. While the model appears compelling, the absence of detectable Yak1 dephosphorylation upon rapamycin treatment is at odds with it [25]. Additionally, as pointed out earlier, the role of Crf1 on RP gene transcription was found to be yeast strain-specific [110] and a potential second mechanism may exist for its regulation. While this does not invalidate the model, it emphasizes existing gaps in our current understanding. The idea of TORC1-dependent PKA localization has also not gained widespread acceptance. In contrast to the findings described above, several other studies detected Tpk1 in the nucleus of glucose-grown cells (in the absence of rapamycin) [148,174,175]. Growth with glycerol as the sole carbon source [174] or into stationary phase [149,175] resulted in partial re-localization to the cytoplasm. Available data on Tpk2 and Tpk3 also lean towards nuclear localization under beneficial and cytoplasmic (possibly P-body or stress granule) localization under nutrient-limited conditions [174,175]. While in these cases cytoplasmic localization corresponds to the inactive state of PKA, Griffioen et al. reported Tpk1 cytoplasmic localization when treating a *cyr1∆* strain with cAMP [149]. A model that rationalizes these differing observations on Tpk1 localization is lacking to date. If TOR signaling indeed activates PKA, it would be expected that nitrogen source or amino acid availability, stimuli that positively regulate TORC1 [176,177,178], also influence PKA activity. Indeed, activation of the PKA target trehalase upon addition of a nitrogen source has been demonstrated and apparently occurs in a cAMP-independent manner [46,179,180]. As this upregulation of trehalase activity additionally requires the presence of a fermentable carbon source, the proposed nitrogen source-dependent, but cAMP-independent PKA activation was referred to as the “fermentable-growth-medium induced pathway”. This concept would be consistent with TOR-dependent regulation of PKA. More recent findings point towards nitrogen source and amino acid-dependent PKA activation via nutrient transceptors and under certain conditions, most notably the addition of L-citrulline to nitrogen-starved cells, TORC1 activity was found dispensable [45,47]. While this is not inconsistent with additional activation by TORC1, it alleviates the need for positing this mechanism.

## 6. Conclusions

There exists compelling evidence for a tight interplay between TOR and PKA signaling, based on genetic findings and the phosphorylation state of known targets. However, genetic evidence also clearly demonstrates that the pathways are not purely epistatic. Based on the best-understood shared targets presented in this review, it is apparent that each point of convergence between the pathways needs to be considered individually. While in some cases the phospho sites targeted by the two pathways are distinct, with TORC1-dependent sites sometimes dephosphorylated by PP2A/PP2A^Sit4^, in other cases the phospho sites are shared between PKA and Sch9 which exhibit similar substrate specificity. The mechanistic basis for some substrates being shared between the two AGC kinases and others being unique to PKA remains an important question to resolve. The review of each of these individual targets also points out that much remains to be learned about how TOR and PKA regulate even these best-understood common targets. In addition, some observations have led to the proposal of a more direct cross-talk between the pathways. While TOR signaling appears like an upstream regulator of PKA based on nucleo-cytoplasmic localization of PKA subunits and on the regulation of Yak1, a better understanding of the mechanism of this proposed cross-talk is required to lend credibility to its existence. Finally, we understand little about the physiological role of the interplay between the two major growth-regulatory pathways. The simple model, according to which PKA solely responds to carbon source availability and TOR to amino acid and nitrogen availability, has increasingly eroded in recent years. Instead, it now becomes important to ask how and why different environmental conditions impact the output of each individual pathway and their interplay. 

## Figures and Tables

**Figure 1 biomolecules-12-00210-f001:**
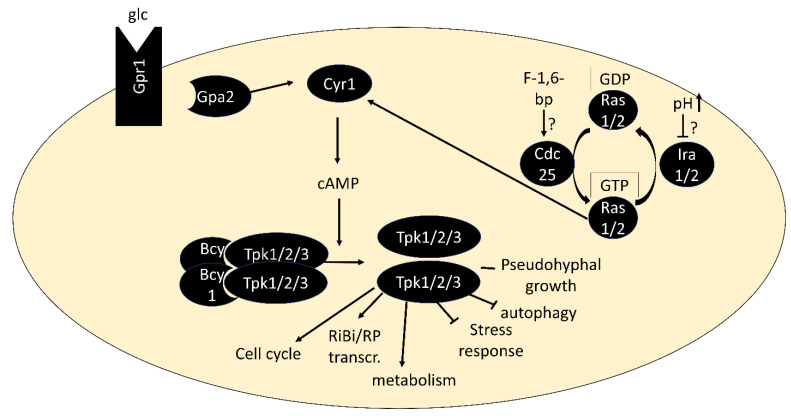
Core components of the PKA pathway. In its inactive form, PKA exists as a tetramer of two regulatory subunits (Bcy1) and two catalytic subunits (Tpk1, Tpk2 or Tpk3). Binding of cAMP to Bcy1 leads to dissociation of the complex and activation of the catalytic subunits. Two main routes of activation of adenylate cyclase Cyr1 exist: via the G protein-coupled receptor Gpr1 and its G protein alpha subunit Gpa2 and via the small GTPases Ras1/2.

**Figure 2 biomolecules-12-00210-f002:**
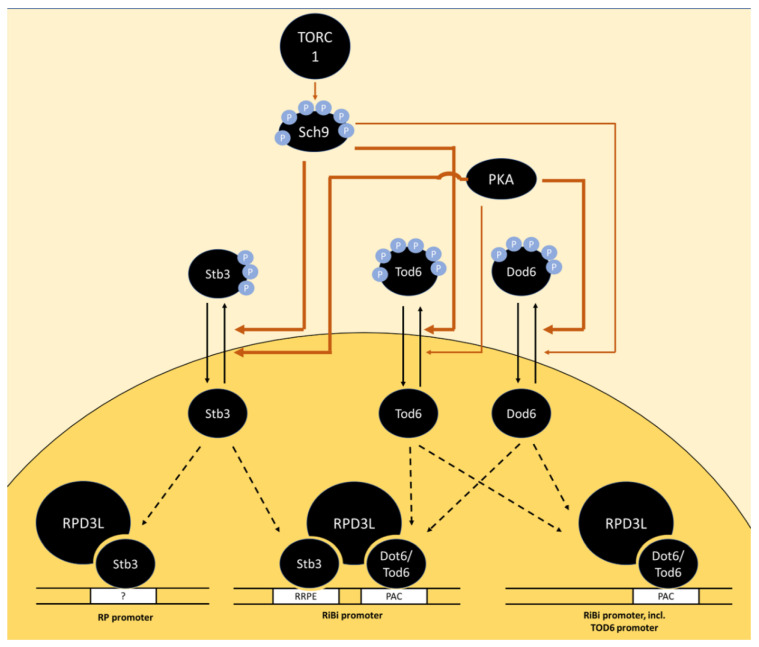
Model of the regulation of RP and RiBi genes via the transcriptional repressors Stb3, Tod6 and Dot6 downstream of TORC1 and PKA. In the absence of phosphorylation by Sch9 and PKA, the transcription factors Stb3, Tod6 and Dot6 bind specific promoter elements upstream of a subset of RP and RiBi genes, triggering their repression via the recruitment of histone deacetylase complex RPD3L. The promoter element bound by Stb3 in RP promoters is marked with a question mark as RP promoters generally do not contain RRPE sequences and the mode of interaction with these promoters is unclear. The thickness of arrows indicates a potentially stronger relative contribution of PKA than Sch9 on Dot6 phosphorylation and vice versa for Tod6 phosphorylation. Potential dephosphorylation of Tod6 by PP2A is omitted as direct evidence is lacking.

**Figure 3 biomolecules-12-00210-f003:**
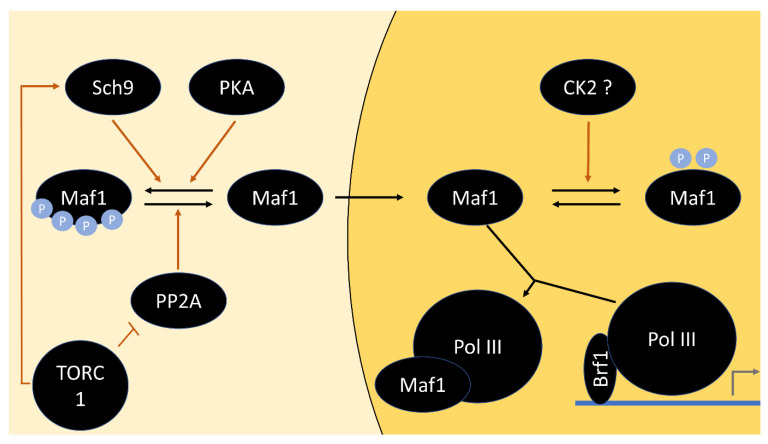
Model of the regulation of Pol III transcription factor Maf1. Maf1 is phosphorylated by Sch9 and PKA to prevent its nuclear localization. This is counteracted by dephosphorylation by PP2A. Unphosphorylated Maf1 translocates to the nucleus and binds Pol III in a manner that prevents association with the transcriptional activator Bfr1, and therefore impedes transcription initiation. Possibly, phosphorylation of Maf1 on sites other than the PKA and Sch9 sites, potentially by CK2, hinders Maf1 association with Pol III. The subcellular locations at which Maf1 phosphorylation and dephosphorylation take place are unknown and may differ from the ones shown).

**Figure 4 biomolecules-12-00210-f004:**
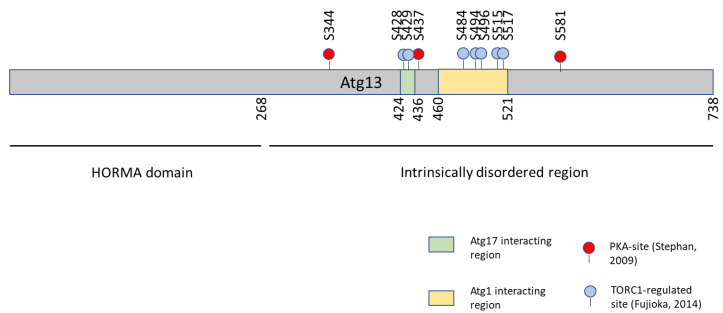
PKA- and TORC1-regulated phosphorylation sites on Atg13. Sites reported by Stephan, 2009 are shown in red and by Fujioka, 2014 in blue. The position of the Atg17-interacting region is indicated by a green and the Atg1-interacting region as a yellow rectangle.

**Figure 5 biomolecules-12-00210-f005:**
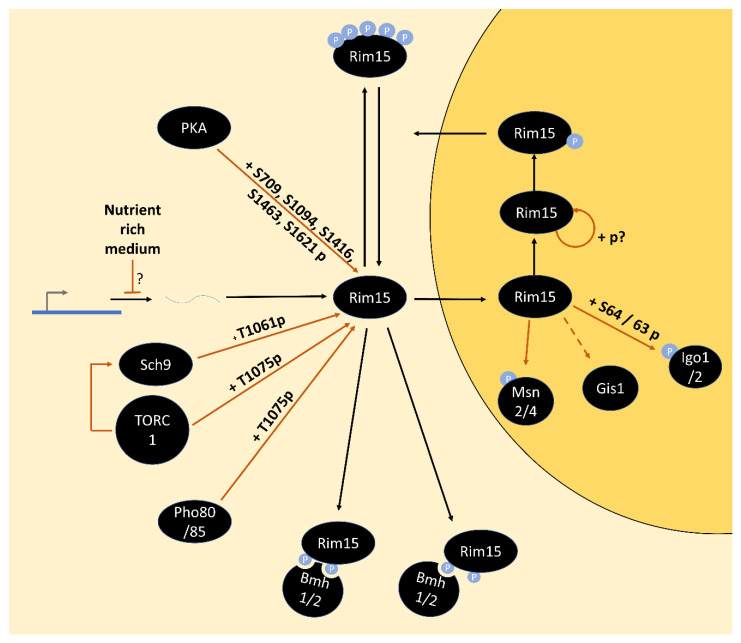
Model for signaling through Rim15. Rim15 is phosphorylated on T1062 by Sch9 and on T1075 by TORC1 and Pho80/85. Phosphorylation of these sites leads to cytoplasmic retention by Bmh1/2. Phosphorylation on five other sites by PKA inactivates Rim15 catalytic activity. This PKA-dependent phosphorylation may be limited to the cytoplasm due to association of PKA catalytic with regulatory subunits in the nucleus. Rim15 autophosphorylation may be required for its nuclear export. A form of Rim15 phosphorylated by both PKA and Sch9/TORC1/Pho80/85 is omitted for clarity. Phosphorylation of substrates by Rim15 is only shown in the nucleus for simplicity, while the true location has not yet been determined.

**Figure 6 biomolecules-12-00210-f006:**
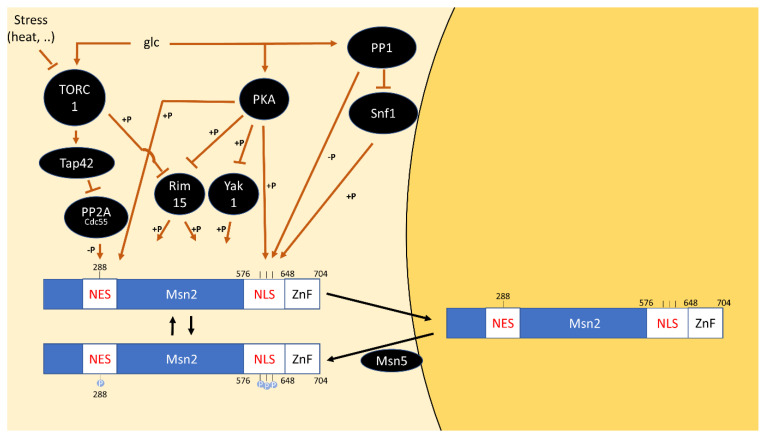
Model for the phospho-regulation of Msn2. Both the nuclear export signal (NES) and nuclear localization signal (NLS) are phosphorylated by PKA to promote Msn2 cytoplasmic localization. The NES is presumably dephosphorylated by PP2A^Cdc55^ upon TORC1 inactivation. The NLS is additionally phosphorylated by Snf1 and dephosphorylated by PP1. The localization of phosphorylation sites by Rim15 and Yak1 on Msn2 are unknown. Nuclear-cytoplasmic localization as a consequence of NES and NLS phosphorylation is shown, but the subcellular compartment in which phosphorylation events take place is unknown. A speculative additional dephosphorylation of an unknown site in the zinc-finger domain (ZnF) by PP2A^Cdc55^ is omitted.

**Figure 7 biomolecules-12-00210-f007:**
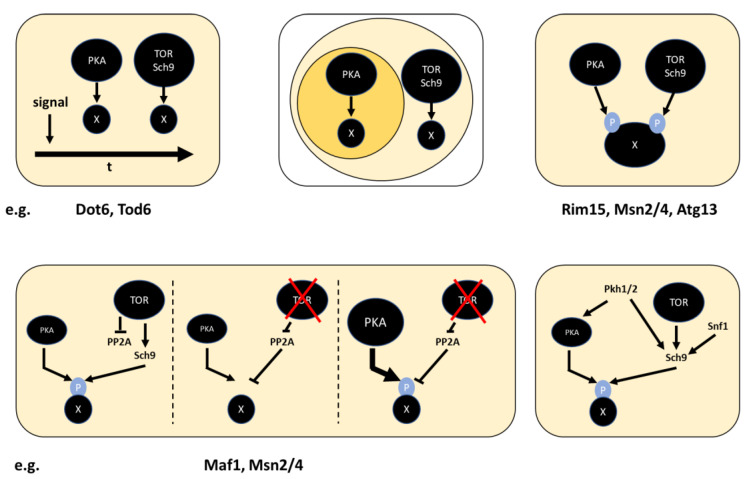
Speculative mechanisms of interaction of PKA and TOR/Sch9 on shared targets. Top: Differential temporal (left) and subcellular (middle) activities of the pathways. Right: Phosphorylation of different sites by the two pathways. Bottom, left: Control of substrate phosphorylation by TOR via activation of Sch9 and inactivation of PP2A. PP2A may overcome weak, but not strong PKA activity. Right: Regulation of Sch9 via pathways additional to TOR. Activation by Pkh1/2 is shared with PKA. Substrates for which the mode of interaction may be relevant are proposed.

**Table 1 biomolecules-12-00210-t001:** Genetic interactions of PKA and TOR signaling.

Observed Phenotype	Reference	Interaction Type *
*bcy1Δ* rescues growth defect of *sch9Δ*	Toda, 1988 [70]	TOR + PKA
*SCH9*-overexpression rescues inviability of *tpk1Δ tpk2Δ tpk3Δ*, *ras1Δ ras2Δ* and *cyr1Δ*	Toda, 1988 [70]	PKA + TOR
*SCH9* rescues temperature-sensitivity of *cdc25-ts*	Toda, 1988 [70]	PKA + TOR
*ras2^V19^-*, *CDC25*- or *TPK1*-overexpression increase rapamycin resistance of *gat1Δ gln3Δ*	Schmelzle, 2004 [71]	TOR + PKA
*bcy1Δ* increases rapamycin resistance of *gat1Δ gln3Δ*	Schmelzle, 2004 [71]	TOR + PKA
*ira2Δ, bcy1Δ* and *ras^V19^* mutations increase rapamycin resistance	Zurita-Martinez, 2005 [72]	TOR + PKA
*ras2Δ*, *tpk1Δ*, *tpk2Δ* and *tpk3Δ* increase rapamycin sensitivity	Zurita-Martinez, 2005 [72]	TOR + PKA
*tpk1Δ tpk2Δ tpk3Δ yak1Δ* and *tpk1Δ tpk2Δ tpk3Δ msn2Δ msn4Δ* increase rapamycin sensitivity	Zurita-Martinez, 2005 [72]	TOR AND PKA
*BCY1* and *PDE2* overexpression rescue temperature sensitivity of *kog1-ts*	Araki, 2005 [73]	TOR -PKA
*ras1Δ ras2-23* mutations increase rapamycin resistance (1.5–2.5 ng/mL)	Ramachandran, 2011 [74]	TOR -PKA
*bcy1Δ* increases rapamycin sensitivity (1.5 ng/mL)	Ramachandran, 2011 [74]	TOR -PKA
Expression of *ras^V19^* causes rapamycin sensitivity (3 ng/mL)	Ramachandran, 2011 [74]	TOR -PKA
Overexpression of *PDE2* causes rapamycin resistance (3 ng/mL) and rescues temperature-sensitivity of *tor2-ts*	Ramachandran, 2011 [74]	TOR -PKA
*ras^V19^* mutant shows synthetic growth defect with *tor1Δ* and *tor1Δ tor2-ts* and with *tor2-ts* at non-permissive temperature	Ramachandran, 2011 [74]	TOR -PKA
Rapamycin treatment increases phosphorylation of PKA targets Srb9, Rim15 (after 2 h) and Cki1 (~2–3 h)	Ramachandran, 2011 [74]	TOR -| PKA
*sch9Δ* has increased basal trehalase activity during growth on glycerol, but magnitude of increase after glucose addition is decreased	Crauwels, 1997 [75]	TOR -| PKA; TOR -> PKA

* The reported interaction is consistent with TOR + PKA: positive interaction w. possible epistasis of TOR over PKA. PKA + TOR: positive interaction w. possible epistasis of PKA over TOR. TOR AND PKA: positive interaction via AND gate. TOR -PKA: negative interaction. TOR -| PKA: negative interaction: TOR represses PKA. TOR -> PKA: positive interaction: TOR activates PKA.

## Data Availability

Not applicable.

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
