# Peer review of "Interaction of TOR and PKA Signaling in S. cerevisiae"

_biomolecules, 2022, doi:10.3390/biom12020210_

Round 1

Reviewer 1 Report

With this manuscript, Michael Plank provides probably the most comprehensive review on downstream targets of TORC1 and PKA pathways in yeast to date.

Starting with describing the extensive genetic interactions found for both signaling pathways, which originally have led to the proposal of different epistatic relationships (TOR upstream of PKA, or TOR and PKA parallel), he focusses on discussing important downstream targets and how these are regulated by TOR and PKA, respectively. By large, the data are well consistent with TOR and PKA acting in parallel on the same targets, sometimes even on the same residues.

The review is exceptionally comprehensive, which makes it more of an interesting resource rather than an interesting read beyond a very specialized field of yeast researchers.

Nevertheless, one key argument for the parallel nature of PKA and TOR signaling cascades was the differential responsiveness to nitrogen and carbon sources. While in lines 805ff and 828ff Nitrogen regulation of PKA signaling is mentioned as a potential argument for direct crosstalk, the data for these claims are not described in any detail and only one citation is given.

Also, to open the review to a more broader readership, I wonder if the review would benefit from highlighting – as much as possible – the more general interest of studying the convergence of studying the interplay of two growth promoting pathways on different targets in S.cerevisiae.

The argument that PKA activity peaks soon after glucose readdition in parallel to the cAMP pulse (lines 579ff) seems not substantiated by time course analysis of PKA-dependent phosphorylation sites, or localization of stress sensitive transcription factors, and only one citation (55, a review) is given in support of “peak PKA activity”. This argument needs to be weakened considerably, or more data needs to be given in support of “peak PKA activity”.

Minor comments:

Line 57: “Tap42 temperature-, but also rapamycin sensitive” should rather read rapamycin insensitive.

Line 118: is the described cAMP independent regulation also PKA independent? Please clarify

Author Response

I thank the reviewers for their time and suggestions. My responses addressing their comments are in red below.

-Nevertheless, one key argument for the parallel nature of PKA and TOR signaling cascades was the differential responsiveness to nitrogen and carbon sources. While in lines 805ff and 828ff Nitrogen regulation of PKA signaling is mentioned as a potential argument for direct crosstalk, the data for these claims are not described in any detail and only one citation is given.

I deliberately largely excluded addressing of the upstream regulation of TOR and PKA as this would warrant a review in its own right, but I agree that there is a benefit of briefly discussing nitrogen source-dependent regulation. I added a section around lines 824ff.

-Also, to open the review to a more broader readership, I wonder if the review would benefit from highlighting – as much as possible – the more general interest of studying the convergence of studying the interplay of two growth promoting pathways on different targets in S. cerevisiae.

Presenting the TOR-PKA-interplay as an example that will be helpful for researchers thinking about signaling interplay in different contexts is indeed a major aim of this review. On the other hand, discussing specific similarities in other pathways would exceed the limits of available space and my area of expertise. I added statements highlighting the relevance in a broader context at key sections of the manuscript (The introduction and “3.4. Summary of shared PKA and TOR targets.”). Beyond this, I expect readers will be able to identify parallels in their system of interest.

-The argument that PKA activity peaks soon after glucose readdition in parallel to the cAMP pulse (lines 579ff) seems not substantiated by time course analysis of PKA-dependent phosphorylation sites, or localization of stress sensitive transcription factors, and only one citation (55, a review) is given in support of “peak PKA activity”. This argument needs to be weakened considerably, or more data needs to be given in support of “peak PKA activity”.

I agree that this statement was based on the inherent assumption that PKA activity follows cAMP levels. I weakened the original statement by spelling out this assumption (line 590ff). I also agree with the stated arguments against this assumption, while the results in ref. 83 argue in its favor. The probability of the assumption being true is however not critical here, as it is only brought up as an example for possible temporal changes in PKA activity.

Minor comments:

-Line 57: “Tap42 temperature-, but also rapamycin sensitive” should rather read rapamycin insensitive.

 Thank you for spotting this error. Corrected.

-Line 118: is the described cAMP independent regulation also PKA independent? Please clarify

I split the sentence into two and clarified the cAMP independence. I also highlighted the ambiguity between PKA-independence vs cAMP-independent PKA activity.

Reviewer 2 Report

This is an interesting, thorough review in the interaction of TOR and PKA pathways. It provides a detailed up to date report of the current state of knowledge and raises pertinent questions that remain to be addressed in the field.

I only have a few minor suggestions.

Table 1: some sentences carry a full stop, others do not.

Genes and mutations should be italicized. In fact, this issue should be addressed throughout the text.

In Fig. 1, doing a small enlargement of the font would benefit the image. Likewise, for the phosphoresidues in Fig. 5.

Line 453- section title misplaced

In pag. 14 reference to the work “TOR and PKA signaling pathways converge on the protein kinase Rim15 to control entry into G0” by Pedruzzi et al 2003 Mol Cell. would complete the discussion.

Author Response

I thank the reviewers for their time and suggestions. My responses addressing their comments are in red below.

-Table 1: some sentences carry a full stop, others do not.

The remaining full stops were removed.

-Genes and mutations should be italicized. In fact, this issue should be addressed throughout the text.

Genes and mutations were italicized.

-In Fig. 1, doing a small enlargement of the font would benefit the image. Likewise, for the phosphoresidues in Fig. 5.

The fonts on both figures have been enlarged.

-Line 453- section title misplaced

The misplacement was corrected.

-In pag. 14 reference to the work “TOR and PKA signaling pathways converge on the protein kinase Rim15 to control entry into G0” by Pedruzzi et al 2003 Mol Cell. would complete the discussion.

I agree, this is an important reference in this context. It has been added to the relevant section.